# TRPA1-Activated Peptides from Saiga Antelope Horn: Screening, Interaction Mechanism, and Bioactivity

**DOI:** 10.3390/ijms26052119

**Published:** 2025-02-27

**Authors:** Chengwei Wang, Chunjie Wu, Linjiang Song

**Affiliations:** 1State Key Laboratory of Southwestern Chinese Medicine Resources, School of Pharmacy, Chengdu University of Traditional Chinese Medicine, Chengdu 611137, China; chengweiwang@stu.cdutcm.edu.cn; 2Innovative Institute of Chinese Medicine and Pharmacy/Academy for Interdisciplinary, Chengdu University of Traditional Chinese Medicine, Chengdu 611137, China; 3Sichuan Engineering Research Center for Endangered Medicinal Animals, Chengdu 611137, China; 4School of Medical and Life Sciences, Chengdu University of Traditional Chinese Medicine, Chengdu 611137, China

**Keywords:** TRPA1, Saiga antelope horn, peptides, AUF-LC/MS, 5-HT

## Abstract

Saiga antelope horn (SAH), a rare traditional Chinese medicine, exhibits activities of anti-feverish convulsions and anti-inflammation, whereas its underlying mechanism and specific pharmacological components are still unclear. In the present study, transient receptor potential ankyrin 1 (TRPA1), a major transient receptor potential cation channel was used as a target protein to identified TRPA1 high-affinity peptides (THPs) from SAH digests. Firstly, the SAH was digested under in vitro gastrointestinal conditions. With the method of affinity ultrafiltration and liquid chromatography–mass spectrometry (AUF-LC/MS), about 200 peptides that have a high-affinity interaction with the TRPA1 protein were screened from SAH digests. Subsequently, bioactivity databases and molecular docking were further exploited to identified three THPs, including RCWPDCR, FGFDGDF, and WFCEGSF. Furthermore, RIN-14B cells, characterized by the high expression of TRPA1 on cell surfaces, were used as the cell model to investigate the biological effect of THPs. Immunofluorescence and ELISA were conducted and showed that THPs can increase the intracellular Ca^2+^ concentration and serotonin (5-HT) secretion in RIN-14B cells by activating TRPA1, which is evidenced by impaired upregulation of intracellular Ca^2+^ levels and 5-HT secretion after pretreatment with the TRPA1 inhibitor (HC-030031). Moreover, an analysis of Western blots displayed that THPs up-regulated the expression levels of the 5-HT synthesis rate-limiting enzyme (TPH1) and 5-hydroxytryptophan decarboxylase (Ddc), while serotonin reuptake transporter (SERT) levels were down-regulated, suggesting that THPs enhance 5-HT secretion by regulating the 5-HT synthesis pathway. In summary, our findings demonstrate that THPs, which were identified from SAH digest via TRPA1-targeted affinity panning, exhibited the activation of the TRPA1 channel and enhanced 5-HT release in RIN-14B cells.

## 1. Introduction

Horn from the Saiga antelope (*Saiga tatarica* Linnaeus, SAH) has been used in traditional Chinese medicine for the treatment of febrile convulsions, fever, eclampsia, and hemorrhagic disorders for thousands of years [1]. The wild Saiga antelope population has significantly decreased as a result of the SAH trade’s explosive expansion in recent decades, and the usage of SAH is currently prohibited [2]. To rescue the Saiga antelope, which is endangered, and to satisfy medical and industrial demands, researchers and organizations have devoted a great deal of time and energy to the development of substitutes for SAH [3]. It has been discovered in recent years that the degraded peptide fragments of the keratin-like antelope components could serve as a significant structural foundation for the effectiveness of antelope medications. SAH is directly utilized in medicine as a powder, as well as during gastrointestinal digestion; its primary constituent, keratin, is broken down into a wide variety of peptide compounds. Two bioactive peptides found in SAH, YGQL and LTGGF, have been found through studies to potentially help prevent convulsions [4]. Additionally, a number of peptides generated from SAH, such as FVK, FVVLKK, FYY, LADAK, FAAF, FFSK, AYL, TQYK, and ASYL, have shown promise as potential functional substances of SAH in the treatment of hypertension associated with liver yang hyperactivity syndrome [5]. However, its functional ingredients and mode of action, however, are not well understood.

In our previous study, we found that the latency period of febrile seizures (FS) was effectively prolonged, and the seizure grade of FS was reduced in rat pups after SAH pretreatment. The analysis of major neurotransmitters in the brain showed that the levels of TRP and 5-HT, the major neurotransmitter in the brain, were significantly increased after SAH administration. Changes in brain 5-HT levels and its receptors are thought to play a crucial role in the pathogenesis of seizures [6]. We found that the mRNA and protein expression of ER-β and TPH2 were up-regulated in the hippocampus of FS rats pretreated with SAH. The 5-HT molecule can be produced in the brain through the ER-β/TPH2/5-HT pathway, suggesting that SAH may up-regulate the 5-HTergic system by activating the ER-β/TPH2/5-HT signaling pathway [7].

Studies have shown that the activation of the transient receptor potential ankyrin 1 channel (TRPA1) channels is closely related to 5-HT secretion. Sodium oligomannate has been shown to either directly or indirectly enhance calcium entry in enteroendocrine cells by activating sweet taste receptors and TRPA1. This leads to an increase in CCK and 5-HT release, which in turn increases vagal afferent activity. Sodium oligomannate may influence cognitive processes by activating the EC–vagal afferent pathways [8]. Additionally, hydrogen sulfide activates TRPA1 and promotes the release of 5-HT from the epithelioid cells of the chicken thoracic aorta [9]. TRPA1 is a nonselective cation channel that facilitates the influx of calcium upon activation and is widely expressed in sensory neurons as well as non-neuronal cells, including the epithelium and hair cells [10]. It has been observed that mechanical, heat, and cold stimuli can activate TRPA1, and a variety of variables, such as pH, Ca^2+^, trace metals, reactive oxygen species, nitrogen, and carbonyl species, can affect its action. TRPA1 has important roles in the pathophysiology of almost all organ systems, is implicated in inflammation associated with both acute and chronic pain, and is a desirable target for the treatment of disorders connected to it [11]. Mutations in the TRPA1 gene may not only impact sensory nerves and the microvasculature, causing neurological discomfort and vascular issues [12], but may also contribute to ozone-induced lung injury [13]. Wang et al. discovered that TRPA1 can prevent contrast-induced acute kidney injury by controlling mitochondrial fission/fusion, biogenesis, and dysfunction. TRPA1 activation may provide a novel treatment strategy for preventing contrast-induced acute renal damage [14]. Research has demonstrated that the astrocyte-produced TRPA1 protein plays a role in learning, memory, and social interactions, among other functions. Moreover, vagal sensory ganglia can be stimulated using bacterial tryptophan catabolites via EEC TRPA1 signaling [15].

A subset of specialized intestinal epithelial cells (IECs), also known as enterochromaffin cells (ECs), function as chemosensors, detecting a wide range of environmental stimuli such as the gut microbiota, mechanical stimuli, and metabolites, which in turn cause the production and secretion of serotonin (5-hydroxytryptamine [5-HT]). More than 90% of peripheral 5-HT, which is essential for immunological responses, intestinal motility, platelet function, bone growth, and heart function, is produced in vivo by ECs. The disruption of 5-HT, an important intestinal hormone, results in a range of neurological and metabolic disorders [16]. According to earlier studies, chemicals released by 5-HT-containing ECs are expected to act through diffusion onto vagal afferent nerve terminals that project onto the mucosa and subsequently transmit sensory data to the brain [17]. Additionally, through the microbiota–gut–brain axis, intestinal 5-HT release can mitigate chronic stress-induced cognitive impairment in mice [18].

In summary, we hypothesize that a large number of peptides produced by antelope horn after gastrointestinal digestion can exert antipyretic effects by up-regulating the expression of 5-HT through the TRPA1 protein in enterochromaffin cells. However, determining the distinctive peptides of antelope horn with unique potency is a pressing task. TRPA1 high-affinity peptides (THPs) were discovered from the degraded peptide mixture of antelope horn utilizing AUF-LC/MS, bioactivity databases, and molecular docking approaches. Furthermore, the mechanism of THPs in controlling serotonin release from RIN-14B cells was investigated, revealing new information about the mechanism of action of bioactive peptides from SAH in suppressing FS. Furthermore, this work lends scientific support to the creation of artificial alternatives to SAH.

## 2. Results

### 2.1. TRPA1-Binding Peptides Were Selected from SAH Digests by AUF-LC/MS

The raw files generated from the SAH digestion and ultrafiltration mixture after LC-MS/MS data acquisition were opened using Xcalibur to see the following total ion flow chromatograms (Figure 1B,C). PEAKS studio was utilized in this investigation due to its comprehensive scoring model and global optimization methods, which regularly yield more accurate peptide sequences with a higher degree of confidence [19]. Peptide sequences were generated with a de novo score greater than 80%. De novo score values are expressed as a percentage, with higher scores resulting in more accurate results. Free peptides originating from the SAH digestion and ultrafiltration mixture were analyzed using nano LC-MS/MS. Among them, 4659 peptide sequences were obtained from SAH digestion, and 1220 peptide sequences were obtained from the ultrafiltration mixture.

PeptideRanker is a comprehensive bioactive peptide prediction server based on an N-1 neural network, where input peptide sequences are given a score between 0 and 1, and the closer it is to 1, the higher the probability of being biologically active, as well as having fast and high-throughput advantages that enable activity prediction of hundreds of peptides in a short period [20]. The potential bioactivity of the identified peptides was analyzed using PeptideRanker, and peptides with a PeptideRanker score ≥ 0.5 were considered to be potentially bioactive. The prediction results are shown in Figure 1E, whereby 85 peptides with a PeptideRanker score > 0.5 are likely to be biologically active, and 28 peptides with a PeptideRanker score > 0.8 are highly likely to be biologically active (Figure 1E). Peptides with a PeptideRanker score > 0.8 were used for further analysis. In addition to biological activity, toxicity is also an important indicator of whether a peptide has potential for development; therefore, the toxicity of 85 peptides was predicted using ToxinPred [21]. The results showed that 92% of the peptides were non-toxic peptides (Figure 1F). In addition, the content of peptides in the mixture is also an important criterion for assessing their drug ability, and 11 peptides were finally obtained by taking the intersection of peptides with a PeptideRanker score > 0.8, non-toxic peptides, and peptide relative intensities in the top 100 (Figure 1G). These 11 peptides were molecularly docked with TRPA1.

### 2.2. THPs Were Identified from TRPA1-Binding Peptides Using Bioactivity Databases and Molecular Docking

In order to identify TRPA1 high-affinity peptides (THPs), we used bioactivity databases and molecular docking to further screen candidate peptides from the TRPA1-binding peptides. The majority of bioactive peptides derived from natural sources have molecular weights of 1 KD or less, according to several studies [3,22,23,24,25]. Consequently, when screening for bioactive peptides, one of the elements to take into account is the peptides’ molecular weight size. Among the 11 potential TRPA1 ligands, only one peptide, GPSGPQGPSGPLQGP, has a molecular weight of 1 KD or more. In addition, since ADCP can only dock peptides containing 5–20 amino acids [26], we set the ADCP score of FGYY, WYLR, and GPSGPQGPSGPLQGP to zero.

Drug design uses molecular docking analysis as a key technique. It clarifies important interactions with the active site and emphasizes the relationship between biological activity and binding. To determine how these peptides interact with TRPA1, computerized molecular docking studies were carried out. We docked peptides and proteins using ADCP. ADCP, also known as AutoDock CrankPep, is a module inside the AuckDock program dedicated to peptide docking. It combines protein folding techniques, as well as the effective form of rigid receptors as the basis for affinity calculations, and performs operations on peptide folding in the context of the energy landscape of the receptor. To optimize the interaction between the peptide and the receptor protein, ADCP employs Monte Carlo methods to compute the folding of the peptide, producing a docked conformation of the peptide [27].

The eight peptides chosen for this investigation with TRPA1 all had binding energies of less than −8.0 kcal/mol, indicating that the small-molecule ligands had strong binding properties to the receptor proteins [28]. Synthesizing the results of the bioactivity database screening and molecular docking results (Table 1), we selected the top three peptides, RCWPDCR, FGFDGDF, and WFCEGSF, all TRPA1 high-affinity peptides (THPs), to map the docking with TRPA1 proteins and conduct the subsequent analysis (Figure 2). The THPs mainly interact with the active binding site of TRPA1 via conventional hydrogen bonding, Pi–cation, Pi–alkyl, and carbon–hydrogen bonding. RCWPDCR forms hydrogen bonds at Lys1048(A), Lys1052(A), Lys593(B), Gly559(B), Asp1053(B), Tyr1049(B), Glu632(B), and Arg1050(B) (Figure 2A); FGFDGDF at Tyr629(B), Glu632(B), Lys593(B), Lys1048(A), Lys1052(A), and Arg1050(B) (Figure 2B); and WFCEGSF at Arg1050(B), Glu594(B), Lys1052(A), Lys593(B), Lys1046(B), Glu632(B), and Lys591(B). In addition, WFCEGSF formed Pi–sigma and Pi–Pi stacks via TYR-1049(B) (Figure 2C). To sum up, the primary residues of the active components offer a chance for additional theoretical direction in the creation of molecules with improved agonistic and binding properties.

### 2.3. Analysis of Stability Results of THPs-TRPA1 Systems

For each of the three THPs-TRPA1 complexes, 100 ns molecular dynamics simulations were ran to confirm the molecular docking simulation results of THPs and TRPA1. The FGFDGDF-TRPA1 complexes’ RMSD curves tended to stabilize progressively, reaching an ultimate value of roughly 0.4 nm, indicating the systems’ structural stability. The final value was near 0.35 nm, and the RMSDs of the RCWPDCR-TRPA1 complex were marginally lower than those of the other complexes, suggesting that this complex was structurally more stable. In comparison to the other complexes, the WFCEGSF-TRPA1 complex’s RMSD was somewhat greater and eventually neared 0.45 nm, indicating that there might be more conformational variations in this complex (Figure 3A). RMSF analyses identified key flexural regions in THPs and TRPA1 proteins, and the RMSF trends of the three complexes were similar, with large fluctuations around certain residues (e.g., 700–750 and 1050–1100). The RMSFs of the FGFDGDF-TRPA1 and WFCEGSF-TRPA1 complexes fluctuated overall, with RMSF values exceeding 0.6 nm for some residues. The RMSF fluctuations of the RCWPDCR-TRPA1 complex were lower, with an average RMSF of about 0.3 nm, indicating a more compact structure (Figure 3B). Gyrate reflects the overall compactness of the complex. The gyrate of the three THPs-TRPA1 systems eventually stabilized at around 4.5 nm, indicating a more compact complex (Figure 3C). SASA was used to assess the degree of surface exposure of the protein–peptide complexes. The final SASA values of the three THPs-TRPA1 systems were stabilized between 35–45 nm^2^, indicating a smaller exposed surface area with high compactness and stability (Figure 3D). During the 100 ns simulation, the number of hydrogen bonds of the three THPs-TRPA1 complexes ranged from 8–20 to 1–3 (Figure 3E), indicating that the complexes all have strong interactions. The binding free energy data for the THPs-TRPA1 complexes are shown in Table 2.

### 2.4. Effect of THPs on RIN-14B Cell Viability

To verify the safety of the peptides, the CCK-8 kit was used to measure the impact of THPs on the proliferation of RIN-14B cells. We assess the toxicity of THPs in RIN-14B cells for 24 h at 0, 12.5, 25, 50, 100, 150, and 200 μM (Figure 4). At 200 μM of FGFDGD, there was a considerable increase in cell viability as compared to the control. (Figure 4B). At 200 μM of WFCEGSF and RCWPDCR, there was no statistically significant difference in cell viability when compared to the control; however, at this point, cell vitality increased to 104.92% and 101.02%, respectively. (Figure 4A,C). The present results suggest that THPs are not toxic to RIN-14B cells in the concentration range examined. Therefore, high concentrations (200 μM) of THPs were selected for analysis in subsequent experiments.

### 2.5. THPs Promote Ca^2+^ Influx and 5-HT Release in RIN-14B Cells

Research has demonstrated that TRPA1 activation results in calcium inward flow and 5-HT release; as a result, it is thought to be a receptor for 5-HT secretion by ECs [15,29,30,31].

Using fluorescence microscopy, we investigated the effects of THPs and AITC, TRPA1 agonists, on the Ca^2+^ concentration in RIN-14B cells in order to better understand the mechanism of TRPA1-mediated 5-HT release. Higher fluorescence intensity indicates higher intracellular Ca^2+^ concentration. AITC and THPs were found to cause a clear increase in intracellular Ca^2+^ concentration (Figure 5A,B). Furthermore, in a concentration-dependent way, AITC and THPs stimulated the release of 5-HT from RIN-14B. At the 50 μM THP concentration, no statistically significant change in 5-HT release was seen when compared to the control. At 100 μM and 200 μM THP concentrations, 5-HT release was substantially higher than the control. The release of 5-HT was, however, reduced under the treatment with the TRPA1 inhibitor HC-030031. Furthermore, HC-030031 prevented the increase in 5-HT secretion caused by the THPs shown in RIN 14B cells (Figure 5C–E). These findings imply that TRPA1 mediates the increase in Ca^2+^ and 5-HT release brought on by AITC and THPs in the EC cell model RIN-14B.

### 2.6. THPs Increase 5-HT Release by Raising the Expression of TPH1 and Ddc and Decreasing the Expression of SERT

Tryptophan hydroxylase 1 (TPH1) is assumed to be the rate-limiting enzyme of intestinal 5-HT synthesis, which catalyzes the formation of 5-HT from the essential amino acid tryptophan (TRP) [32]. The 5-hydroxytryptophan [33] molecule is initially produced in vivo by TRP and then catalyzed by 5-hydroxytryptophan decarboxylase (Ddc) to form 5-HT [34]. The 5-hydroxytryptamine transporter protein (SERT) is an isotropic sodium-transporter of neurotransmitters [35]. Lower SERT activity would result in less serotonin being reabsorbed, which would raise serotonin levels in the cellular supernatant [36].

We investigated the effects of the TRPA1 agonist AITC, the TRPA1 inhibitor HC-030031, and the THPs on the expression levels of TPH1, SERT, and Ddc, three genes involved in the 5-HT production pathway, using qRT-PCR. The findings demonstrated that following the application of THPs and AITC (30 μM) to RIN-14B cells for 24 h, there was a drop in the mRNA expression levels of SERT and an increase in those of TPH1 and Ddc when compared to the control group. Nevertheless, the results mentioned above were undone when HC-030031 (30 μM) was concurrently pretreated on RIN-14B cells (Figure 6). According to all available data, THPs enhance 5-HT secretion by upregulating TPH1 and Ddc mRNA expression and downregulating SERT mRNA expression.

We also looked into how well THPs worked on proteins involved in 5-HT secretion. Western blot was used to show the expression of TPH1, Ddc, and SERT in each treatment group. (Figure 7A–C). Elevated TPH1 and Ddc levels were observed after the administration of AITC and THPs. However, after the HC-030031 treatment, the expression of TPH1 was significantly downgrade. A similar expression was obtained with Ddc (Figure 7D–F). These results demonstrated that 5-HT production is increased under the THP treatment through increased expression of TPH1 and Ddc. Additionally, we observed that the treatments with AITC and THPs reduced the amount of SERT, which helped to release 5-HT (Figure 7D–F). On the other hand, following 30 μM HC-030031 pretreatment, SERT expression was considerably higher than in the THP group (Figure 7). We show that treatment with THPs active TRPA1 channels, which increase the expression of TPH1 and Ddc and decrease the expression of SERT, improving the release of 5-HT.

## 3. Discussion

SAH has a unique and irreplaceable role in the treatment of clinical diseases. It has been used in Chinese medicine for thousands of years. However, the application of SAH is widely restricted due to bioprotection and ethical reasons, so it is crucial to develop artificial SAH or find effective alternative drugs to SAH. Unfortunately, the mode of action, pharmacological mechanisms, and key markers of SAH on diseases are still unclear, which seriously hinders the research of artificial alternatives to SAH. Therefore, we must explore the possible pharmacological mechanisms of SAH to provide a possible theoretical basis for the study of artificial SAH and the search for the best alternatives. Previous experiments have shown that the up-regulation of the ER-β/TPH2/5-HT pathway may be an effective therapeutic mechanism for SAH in the treatment of FS. Moreover, the activation of TRPA1 channels contributes to 5-HT release. In recent years, it has been found that the degraded peptide fragments of keratin-like components in horned animal medicines may be the important efficacy material basis of horned animal medicines [4]. SAH is directly used as a powder in medicine, and its main component, keratin, is degraded into a large number of peptide compounds after gastrointestinal digestion. It is well known that the special biochemical environment of the gastrointestinal tract, mucus, epithelial permeation, and liver elimination as well as other harsh conditions greatly restricts the peptides from being absorbed into the blood circulation through the intestines. More importantly, it is difficult for most compounds in the blood circulation, including peptides, to penetrate the blood–brain barrier and enter the brain. Therefore, we hypothesized that SAH peptides could activate intestinal TRPA1 channels and increase 5-HT secretion to treat FS.

The active ingredients of SAH are keratin and keratin-related proteins, which account for 42.6% of its total protein. Through sequencing and database comparison, it was shown that the enzymatic breakdown of keratin in antelope horn produced a distinct peptide that may be used to identify the authenticity of SAH [37]. The main components of SAH are proteins and polypeptides. Through the activity of different digestive enzymes in the digestive system, the proteins that are absorbed orally into the gastrointestinal tract are broken down into polypeptides and small-molecule proteins to produce therapeutic effects [38]. Therefore, the degradation of SAH into polypeptides through the use of proteases or appropriate methods is of great significance for the study of the active ingredients of SAH. In this study, artificial gastrointestinal fluid was used to simulate gastrointestinal digestion to extract polypeptide substances from SAH, and a total of 4659 polypeptide sequences were obtained.

Endocrine cells, which store and release hormones and neurotransmitters, are also responsible for controlling some aspects of digestive function, including secretion and motility. Within the digestive system, there are over ten distinct types of endocrine cells, each with a unique distribution pattern [39]. The predominant subset of endocrine cells in the small and large intestines is ECs, which secrete 5-HT. Located on the mucosal surface, these cells react to mechanical stimuli or nutrients like glucose and fatty acids by releasing 5-HT into the intestinal wall [40,41,42]. Different gastrointestinal reactions, including colonic motility and feces, are caused by the released 5-HT coordinating the dynamic balance between the immune system, the neurological system, and the enteroendocrine system [43,44,45]. It was suggested by these data that ECs function as sensors in the mucosa of the gastrointestinal tract. These sensory functions’ underlying cellular and molecular mechanisms, however, remain unclear. Notably, several modulatory effects of peptides on 5-HT secretion have been identified in recent research published in the literature. By acting on the microbiota–gut–brain axis (MGBA), the walnut-derived peptide LPLLR ameliorates cognitive impairment in colitis mice induced by dextran sodium sulfate. At the phylum and genus levels, LPLLR altered the abundance of a diverse array of gut microbiota exhibiting elevated Prevotella and Akkermansia linked to TRP, 5-HT, and 5-hydroxyindoleacetic acid [46]. Three walnut-derived peptides, IPAGTPVYLINR, FQGQLPR, and VVYVLR, have been demonstrated in a study to have strong anti-inflammatory qualities [47]. These peptides attenuated cell death and inflammation while reducing 5-HT, tumor necrosis factor-alpha, and vascular endothelial growth factor expression in lipopolysaccharide-induced inflammation in normal human colon mucosal epithelial NCM460 cells. In summary, peptides regulated 5-HT secretion and receptor expression in ways that are antagonistic to each other. This could be because peptides control the expression of several 5-HT synthesis enzymes at once. Tryptophan hydroxylase (TPH), which catalyzes the conversion of the essential amino acid TRP into 5-HT, is assumed to be the rate-limiting enzyme in 5-HT synthesis [48]. In living things, 5-hydroxytryptophan is first produced by TRP and then catalyzed by 5-hydroxytryptophan decarboxylase (Ddc) to form 5-HT. TPH is involved in the synthesis of 5-HT. It comes in two varieties: TPH1 and TPH2. Remarkably, peripheral 5-HT production is influenced by TPH1, which is mostly expressed by ECs and other non-neuronal cell types (such as adipocytes) [49]. TPH2 influences the synthesis of 5-HT in the central nervous system and is mostly present in the brainstem and mesenteric neuronal cells. Research has demonstrated that antelope horn considerably increases the expression of the TPH2 protein in the nucleus of the median slit and the ER-β/TPH2/5-HT pathway in the hippocampus of febrile seizure rats. This increases the amount of 5-HT in the brain by upregulating TPH2 expression, which in turn prevents the convulsions in FS rats [7]. The 5-hydroxytryptamine transporter protein (SERT) is a neurotransmitter sodium isotropic transporter. Inhibiting the SERT protein can efficiently raise the extracellular 5-HT level [50].

As previously stated, TRPA1 channels are tightly linked to 5-HT secretion [51]. TRPA1 is most often recognized as a chemoreceptor in the body. It can be triggered by a variety of structurally unrelated natural chemicals, including allyl isothiocyanate [52]. TRPA1 ligands can be generically classified into two types: electrophilic and non-electrophilic modulators. Electrophilic agonists activate TRPA1 via a cluster of cysteine residues located at the channel’s N-terminus [53]. TRPA1 ligands can be generically classified into two types: electrophilic and non-electrophilic modulators. Electrophilic agonists activate TRPA1 via a cluster of cysteine residues located at the channel’s N-terminus [53]. The mechanism of activation of non-electrophilic ligands remains unknown. In our investigation, we subjected the mixture formed from in vitro simulated gastrointestinal digestion of SAH to affinity ultrafiltration with the TRPA1 protein and then looked for putative TRPA1 ligands using de novo sequencing and computer analysis. Key active peptides in SAH that bind to TRPA1 were discovered using a bioactivity database and molecular docking study. The peptides with the highest scores were RCWPDCR, FGFDGDF, and WFCEGSF. A search of the literature revealed that these three polypeptides were not found in goat horn or other keratin sources (e.g., skin).

The RIN-14B cell was used as an EC model in this investigation. ECs are electrically excitable cells that oscillate spontaneously with Ca^2+^ [54]. They make up only about 1% of the GIT epithelial cells and are widely scattered. Primary ECs are difficult to isolate and examine. Thus, cell lines generated from enteroendocrine malignancies have been commonly employed in the research of ECs [41,55]. Researchers examined dozens of cell lines from gastrointestinal and endocrine organs, as well as newly isolated ECs, for the expression of EC markers (e.g., TPH1, chromogranin A, VMAT1, and synaptophysin) [29]. They discovered that the rat pancreatic endocrine cell line RIN-14B was the most similar to ECs, since it overexpressed EC marker genes. Previous research has found that, like primary ECs, RIN-14B cells exhibit electrical excitability mediated predominantly by NaV1.3 [56]. Furthermore, RIN-14B cells show spontaneous Ca^2+^ calcium oscillations, which are indicative of primary ECs [54]. These findings indicate that RIN-14B is a reliable model for ECs.

Ca^2+^ ions are important second messengers that regulate the release of hormones and neurotransmitters from vesicles. In our calcium imaging experiments, we found that the application of three antelope peptides induced an enhancement of calcium ion inward flow in RIN-14B cells. Notably, in RIN-14B cells, spontaneous Ca^2+^ oscillations appear to be mediated mostly by TRPA1. According to earlier research, TRPA1 is an excitatory calcium-permeable nonselective cation channel that is crucial for the perception of pain and inflammation of the nervous system [57,58]. TRPA1 is activated by a variety of chemical irritants, microbial metabolites, and specific herbal constituents, such as AITC and other substances derived from food flavorings [51]. Since TRPA1 is highly expressed in ECs, activating it causes 5-HT to be released from RIN-14B cells [29] and natural ECs [54]. It follows that the enhancement of TRPA1-mediated Ca^2+^ inward flow by SAH peptides, which in turn promotes 5-HT release, is not surprising.

We found that SAH peptides were shown to boost 5-HT production in RIN-14B cells by upregulating the expression of TPH1 and Ddc and decreasing the expression of SERT. According to these findings, SAH peptides specifically raised the amount of 5-HT in ECs, which was mediated by TPH1. This study reveals the mechanism by which SAH peptides affect 5-HT secretion, providing a potential therapeutic strategy for peripheral 5-HT dysregulation-related diseases. The research revealed three naturally occurring substances as new TRPA1 agonists and proposed that THPs stimulate 5-HT release by activating the TRPA1 channel in RIN-14B cells. Clarifying the mechanism of TRPA1 activation may be aided by the identification of these compounds. This discovery offers a scientific foundation for SAH therapy resource screening and assessment. This study’s primary peptides and targets also offer a clear path forward for the creation of substitutes for SAH. It is possible to generate new therapeutic compounds that meet clinical needs while simultaneously helping to safeguard endangered species by screening herbs with similar bioactivities or other natural sources.

## 4. Materials and Methods

### 4.1. Materials

Antelope horn powder (Cat no. 210403) was legally purchased from the Chengdu Minjiangyuan company (Chengdu, China); artificial gastric fluid (Cat no. SL6600) and artificial small intestine fluid (Cat no. SL6610A) were purchased from coolaber (Beijing, China); recombinant rat transient receptor potential cation channel subfamily A member 1 (TRPA1) and partial (Cat no. CSB-EP764869RA) were obtained from Cusabio (Wuhan, China); allyl isothiocyanate, a TRPA1 agonist (Cat no. 57-06-7), was purchased from Aladdin (Shanghai, China); HC-030031 (Cat no. HY-15064) was obtained from MedchemExpress (Princeton, NJ, USA); Dulbecco’s modified Eagle medium (Cat no. 8123621) was obtained from Gibco Inc. (Saranac, NY, USA); fetal bovine serum (Catalog No. FSP500) was obtained from Excell (Suzhou, China); a penicillin–streptomycin–amphotericin mixture (Catalog No. E607064-0100) was from Sangon Biotech (Shanghai, China); and Cell Counting Kit-8 (CCK-8) (Catalog No. BS350E) was purchased from Biosharp (Hefei, China); Fluo-4 AM eater (Catalog No. F3013S) was from UElandy (Suzhou, China); A cell total RNA isolation kit (RE-03111) and RT EasyTM II (RT-01022) were obtain from FOREGENE Biotech (Chengdu, China). Enhanced RIPA lysate (Catalog No. AR 0102), the broad-spectrum protease inhibitor mixture (Catalog No. AR 1182), broad-spectrum phosphatase inhibitor mixtures (Catalog No. AR 1183), Western-specific primary antibody and secondary antibody diluents (Catalog No. AR 1017), and the TBST buffer (Cat no. BL315B) were obtained from Boster Biological Technology (Wuhan, China); The ST/5-HT (Serotonin/5-Hydroxytryptamine) ELISA kit (Cat no. E-EL-0033) was purchased from Elabscience (Wuhan, China); the polyvinylidene fluoride membrane (Cat no. IPVH00010) was purchased from Millipore (Bedford, MA, USA); the tryptophan hydroxylase 1 (TPH1) recombinant rabbit monoclonal antibody (Cat no. ET1610-37), serotonin transporter rabbit polyclonal antibody (ER1916-44), DOPA decarboxylase recombinant rabbit monoclonal antibody (ET1704-94), and GAPDH recombinant rabbit monoclonal antibody (ET1601-4) were purchased from HUABIO (Hangzhou, China); HRP-conjugated goat anti-Rabbit IgG (H+L) (AS014) was purchased from ABclonal (Wuhan, China); The UltraSignal ultrasensitive ECL chemiluminescent substrate was purchased from Beijing 4A Biotech Co. (Beijing, China).

### 4.2. Preparation of Antelope Horn Peptides

The modified reported method served as the basis for the simulation of gastrointestinal digestion [3]. A total of 36 mL of the artificial gastric solution was mixed with 240 mg of antelope horn powder, and the reaction was run for two hours at 37 °C and 200 rpm. The process was then completed to produce the gastric digesting solution, and the pH was raised to 7.0 using 10% NaOH to inactivate the pepsin. For every hour of digestion, a 6 mL sample was obtained, and the enzyme was inactivated for 5 min at 100 °C. After preheating 24 mL of the artificial small intestine solution for 30 min at 37 °C, it was added to the stomach digestive solution. For three hours, a shaking reaction was carried out at 37 °C and 200 r/min. Six milliliters of the sample was taken every hour of digestion, and the enzyme was inactivated for five min at 100 °C. The remaining reactant was then heated to the boiling point of water for five min to deactivate the enzyme, complete the reaction, and yield the pancreatic digestive solution. In order to obtain the tryptic digestive solution, the reaction was completed. Every hour, the digests were combined and centrifuged at 10,000× *g* for 20 min at 4 °C. The supernatant was then ultrafiltered through a 3 kDa ultrafiltration tube (UFC9003, 3 kDa, 15 mL) at room temperature for 60 min, with the filter membrane facing upward. The resulting ultrafiltrate was then lyophilized and kept at −20 °C for further analysis and activity tests.

### 4.3. Ultrafiltration Screening of Potential TRPA1 Ligands

PBS was used to dissolve the lyophilized powder of the SAH ultrafiltrate. Next, TRPA1 (4 μg, 100 μL) was incubated with 100 μL of the test samples for 30 min at 37 °C. The mixture was then incubated for a further 10 min at 13,000 rpm before being transferred to an ultrafiltration centrifuge tube (UFC5010, 10 kDa, 500 μL). A total of 400 microliters of PBS was added, allowed to settle for a few minutes, and then the unbound compounds were removed using three separate centrifugations at 13,000 rpm for ten min. In order to release the bound ligand, 200 μL of 90% methanol–H_2_O was added to the filter membrane, centrifuged at 10,000× *g* for 10 min at high speed, and repeated three times; the eluent was collected and freeze-dried.

### 4.4. Peptide Characterization by Nano-LC-MS/MS

#### 4.4.1. Sample Pretreatment

After reductive alkylation and desalting the ultrafiltrate lyophilized powder sample, the solvent was removed in a vacuum centrifugal concentrator set at 45 °C. Once again, the sample disintegrated in pristine water.

#### 4.4.2. Liquid Chromatographic Conditions

The following are the liquid chromatographic conditions: analytical column: 100 μm i.d. × 180 mm; packing: Reprosil-Pur 120 C18-AQ 3 μm; mobile phase A: 0.1% formic acid aqueous solution; mobile phase B: 80% ACN/0.1% formic acid aqueous solution; flow rate: 600 nL/min; and analytical time for each component: 66 min.

#### 4.4.3. Mass Spectrometric Conditions

Primary mass spectrometry parameters: resolution: 70,000; AGC target: 3e6; MaximumIT: 100 ms; and scan range: 100–1500 *m*/*z*.

Secondary MS parameters: resolution: 17,500; AGCtarget: 1e5; MaximumIT: 50 ms; TopN: 20; and NCE/steppedNCE: 28.

#### 4.4.4. Database Search

The mass spectrometry raw file was subjected to peptide sequence resolution using the de novo method with the software PEAKS Studio (10.6), and the following parameters were searched: fixed modifications: carbamidomethyl (C); variable modifications: oxidation (M), acetylation (N-term); enzyme: nonspecific; maximum missed cleavages: 2; peptide mass tolerance: 20 ppm; fragment mass tolerance: 20 ppm; and fragment mass tolerance: 0.02 Da.

### 4.5. Prediction of Antelope Horn Peptide Activity

PeptideRanker was used to predict peptide function and biological activity (http://distilldeep.ucd.ie/PeptideRanker/), accessed on 20 April 2024, and peptides with scores greater than 0.5 were considered to be potentially biologically active [59]. Toxicity prediction of peptides was performed using ToxinPred (http://crdd.osdd.net/raghava/toxinpred/) [60], accessed on 20 April 2024. The physicochemical properties of the peptides, including hydrophobicity, hydrophilicity, charge, and molecular weight (MW) were determined using PlifePred (https://webs.iiitd.edu.in/raghava/plifepred/batch.php) [61], accessed on 20 April 2024.

### 4.6. Molecular Docking

To elucidate possible activities, the TRPA1 and the SAH peptides were molecularly docked. TRPA1’s structure is available for download at https://www.rcsb.org/ (ID number: 6V9W), accessed on 22 April 2024. Small molecules and water of crystallization were eliminated to create the crystal structure, which was then stored in the PDB format. To automatically assign atom types and charges, receptor proteins were fed into AutoDock CrankPep (ADCP) and stored in the pdbqt format. The following describes the procedure: Create 3D structures of potential SAH peptides using PyMOL. Then, choose the optimal conformation to use as the structure and store it in the PDB format. Using ADCP, docking and conformations were carried out. The top three affinity peptides were carefully chosen as docking conformations, and PyMOL and LigPlot+ were used for visualization and analysis [62].

### 4.7. Molecular Dynamics Simulation

The Gromacs2022.3 software was used for molecular dynamics simulation [63]. For small-molecule preprocessing, AmberTools22 was used to add the GAFF force field to small molecules, while Gaussian 16W was used to hydrogenate small molecules and calculate the RESP potential. Potential data will be added to the topology file of the molecular dynamics system. The simulation conditions were carried out at a static temperature of 300K and at atmospheric pressure (1 Bar). Amber99sb-ildn was used as a force field, and water molecules were used as the solvent (Tip3p water model), and the total charge of the simulation system was neutralized by adding an appropriate number of Na^+^ ions. The simulation system adopts the steepest descent method to minimize the energy and then carries out the isothermal isovolumic ensemble (NVT) equilibrium and isothermal isobaric ensemble (NPT) equilibrium for 100,000 steps, respectively, with a coupling constant of 0.1 ps and a duration of 100 ps. Finally, free molecular dynamics simulation was performed. The process consisted of 5,000,000 steps; the step length was 2 fs, and the total duration was 100 ns. After the simulation was completed, the built-in tool of the software was used to analyze the trajectory, and the root-mean-square variance (RMSD), root-mean-square fluctuation (RMSF), and protein rotation radius of each amino acid trajectory were calculated and combined with the free energy (MMGBSA), free energy topography, and other data [64].

### 4.8. Cell Culture and Drug Screening

The rat pancreatic islet cell line RIN-14B was obtained from Shanghai Cell Bank (Shanghai, China) and cultured in Dulbecco’s modified Eagle medium with 10% (*v*/*v*) fetal bovine serum and 1% (*v*/*v*) penicillin–streptomycin–amphotericin. Cells were incubated with 5% CO_2_ at 37 °C and passaged at a cell growth density of 80–90%. Cells were inoculated in 96-well plates (2 × 10^4^ cells/well) and incubated for 24 h. Cells were incubated with different concentrations of THPs for 24 h to select the appropriate concentration.

### 4.9. Cell Viability Assays

The toxicity of THPs against RIN-14B cells was assessed using CCK-8. After being plated in 96-well plates with 2 × 10^4^ cells per well, RIN-14B cells were grown at 37 °C for 24 h. RIN-14B cells were incubated for 24 h at various TSBP doses, and cell viability was assessed to determine the proper concentrations. The 5-HT release of 50 μM, 100 μM, and 200 μM THPs; a 30 μM allyl isothiocyanate (AITC)-TRPA1 agonist [65]; and a 30 μM HC-030031-TRPA1 inhibitor [66] was assessed in light of the aforementioned experimental findings.

### 4.10. Measurement of Intracellular Ca^2+^ Concentration

In order to evaluate the Ca^2+^ content in RIN-14B cells, the cells were inoculated onto 12-well plates. Once 50% confluence was reached, 5 μM Fluo-4, AM ester prepared in Ca^2+^-free HBSS was added to the cells along with a suitable amount of a 20% Pluronic F-127 solution. This prevented Fluo-4, AM ester from aggregating in the buffer and encouraged Fluo-4, AM ester to enter the cells; the final concentration of Pluronic F-127 was controlled at 0.04–0.05%. For one hour, the cells were incubated at 37 °C in the dark. They were then cleaned three times using either HBSS or Ca^2+^-free PBS. An additional 80 µL of the HEPES buffer containing Ca^2+^ was added. The test was run on a machine (Fluo-4, AM ester bound to Ca^2+^ with excitation wavelengths of 494 nm/516 nm) as soon as the successive administration was completed. Fluorescence microscopy was used to identify Fluo-4’s fluorescence. Ultimately, the ImageJ 1.8.0 software was used for the processing and analysis of the fluorescence results.

### 4.11. 5-HT Release Measurement

Logarithmic growth phase RIN-14B cells were obtained and plated in 12-well culture plates at a density of approximately 2 × 10^5^ cells per well. The plates were then incubated for the entire night at 37 °C with 5% CO_2_ and then divided into a solvent group (without cells), a blank group (with cells), an agonist group (AITC) (30 μM), an inhibitor group (HC-030031) (30 μM), an antelope horn peptide low group (50 μM), a medium group (100 μM), a high-dose (200 μM) group, and a peptide high-dose group (200 μM)+inhibitor group (30 μM); each group was set up with 3 duplicate wells; supernatants were removed, and 0.5 mL of the complete medium was added to the solvent group (without cells); 0.5 mL of the complete medium was added to the blank group, and 0.5 mL of the complete culture medium containing THPs according to the abovementioned groups was added the drug-dosing group, and the culture was continued for 24 h at 37 °C and 5% CO_2_ in an incubator.

The RIN-14B cell culture supernatant’s 5-HT concentration was determined using an ELISA kit in accordance with the manufacturer’s instructions after the cell supernatants were collected into sterile Ep tubes and centrifuged for 10 min at 2–4 °C. Absorbance was measured at 450 nm using an enzyme meter.

### 4.12. RNA Extraction and Real-Time PCR

Using a cell total RNA isolation kit, total RNA was extracted from RIN-14B cells following treatment. RT EasyTM II was used to transcribe isolated RNA into cDNA. Following 40 cycles of annealing and elongation at 72 °C for 20 s and denaturation at 95 °C for 10 s, the synthesized cDNA was used to perform gene expression analysis via quantitative real-time polymerase chain reaction (qRT-PCR) using a SYBR green-based technique at 95 °C for 3 min. With the 2^−△△ct^ technique, the relative change in gene expression was analyzed [67]. The GAPDH gene was used to standardize the ct data, and the results were represented as fold changes relative to the control. Table 3 contains the list of primers.

### 4.13. Western Blotting

Initially, protein lysates were produced using a loading buffer, boiled for 15 min, and then centrifuged for 10 min at 12,000× *g* at 4 °C. Following their electrophoresis using 10% SDS, they were put onto membranes made of polyvinylidene fluoride. Membranes were blocked for 40 min at 4 °C using the protein-free quick sealing solution; then, they were incubated with primary antibodies for an additional night. After that, membranes were treated for two hours at room temperature with secondary antibodies coupled with horseradish peroxidase. Lastly, the ImageJ 1.8.0 software was used for the analysis and processing of the Western blot results.

### 4.14. Statistical Analysis

The data were analyzed using IBM SPSS Statistics 27 and are displayed in Origin 2024 as the mean ± SD. One-way ANOVAs were used in our investigation to assess statistical significance. Statistical significance was considered as a *p*-value less than 0.05.

## 5. Conclusions

In this study, target protein-based virtual screening and in vitro experiments were used to identify naturally occurring chemicals that activate TRPA1 channels. Using affinity ultrafiltration, liquid chromatography, mass spectrometry, and de novo sequencing, agonist candidates were first chosen from SAH. Then, using the bioactivity database and molecular docking tools, three peptides exhibiting TRPA1 protein activating activity, RCWPDCR, FGFDGDF, and WFCEGSF, were selected from the digested products. Cellular investigations verified that they up-regulated the expression of TPH1 and Ddc and down-regulated the expression of SERT to boost 5-HT production, thereby causing Ca^2+^ inward flow through the activation of the TRPA1 channel. The identification of these peptides may contribute to the understanding of TRPA1’s activation mechanism and offer a clear path forward for the creation of alternatives to antelope horn.

## Figures and Tables

**Figure 1 ijms-26-02119-f001:**
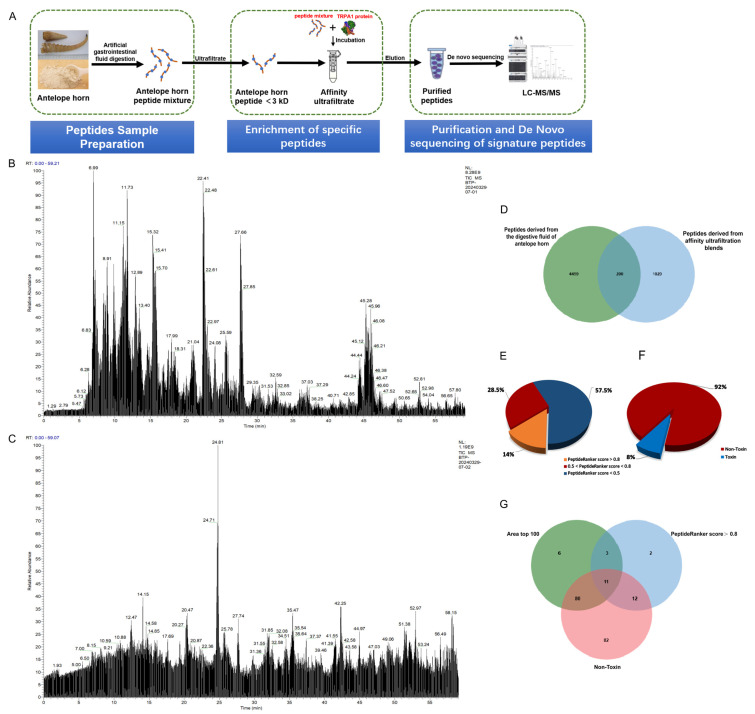
TRPA1-binding peptides were selected from SAH digests. (**A**) Schematic diagram of screening of TRPA1 ligands from SAH digests. (**B**) LC-MS/MS total ion flow chromatograms of SAH digestion. (**C**) LC-MS/MS total ion flow chromatograms of the ultrafiltration mixture of SAH incubated with active TRPA1. (**D**) Venn diagram of the same peptides in SAH digest and the ultrafiltration mixture. (**E**) The outcome of the screening of 200 SAH peptides from PeptideRanker and their (**F**) toxicity. (**G**) Venn diagram of a PeptideRanker score > 0.8, non-toxic peptides, and peptide relative intensities in the top 100.

**Figure 2 ijms-26-02119-f002:**
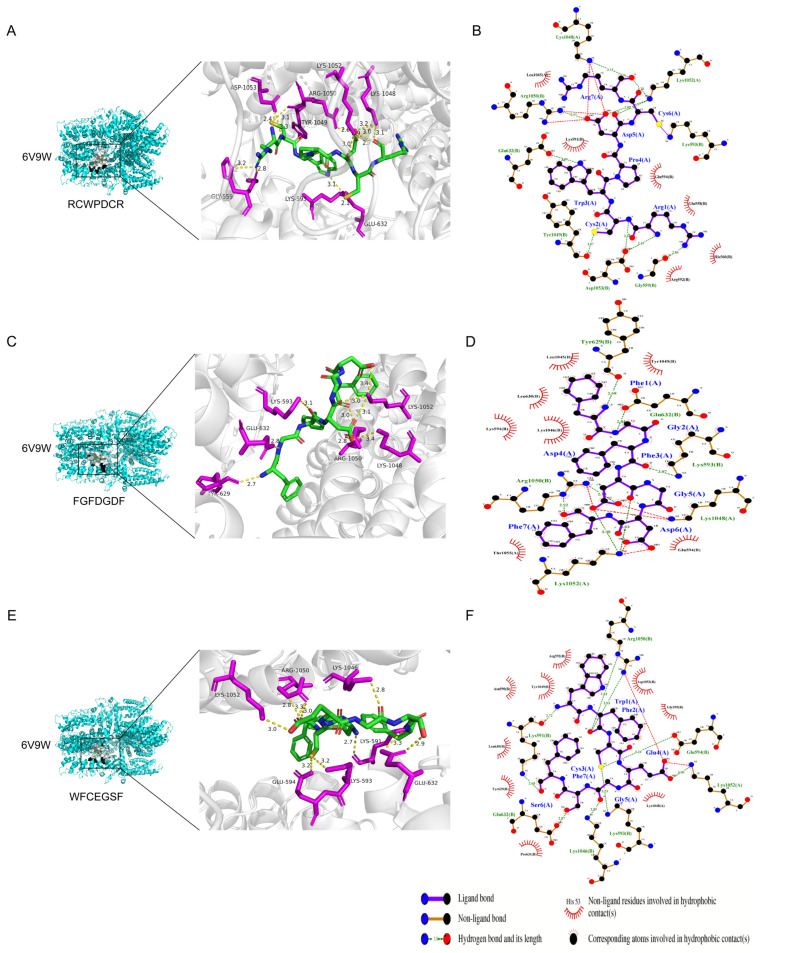
The conformations of the screened SAH peptides docked with TRPA1 via ADCP: RCWPDCR (**A**), FGFDGDF (**C**), and WFCEGSF (**E**). The interaction between THPs and protein residues ((**B**,**D**,**F**) represent the results of RCWPDCR, FGFDGDF, and WFCEGSF binding to TRPA1, respectively).

**Figure 3 ijms-26-02119-f003:**
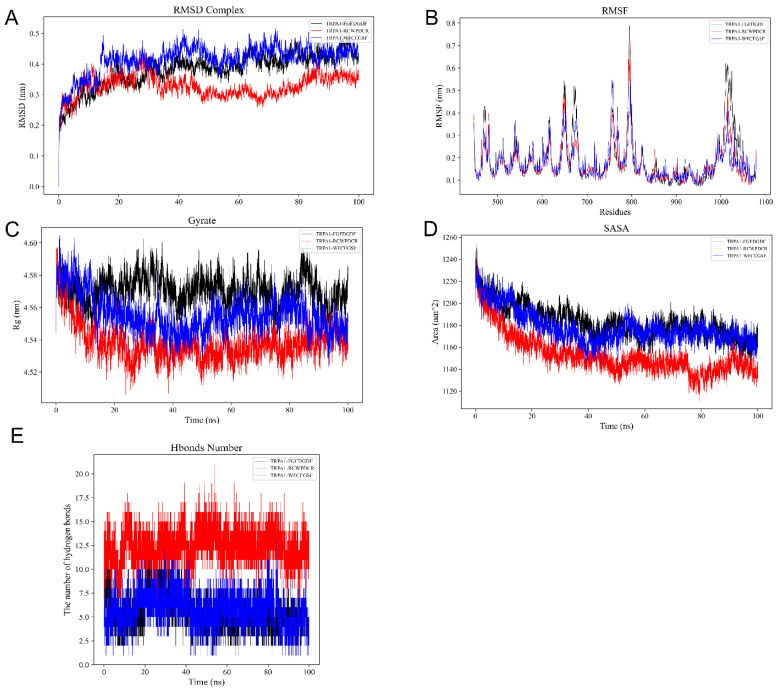
Analysis of stability results of THPs-TRPA1 systems via molecular dynamics simulation. (**A**) The RMSD of the complexes over time during the molecular dynamics simulation. (**B**) The RMSF was calculated from the molecular dynamics simulation trajectory. (**C**) Radius of gyration (Rg) analysis. (**D**) Solvent accessible surface area (SASA) analysis. (**E**) The number of hydrogen bonds between proteins and small molecules changes over the course of the molecular dynamics simulation.

**Figure 4 ijms-26-02119-f004:**
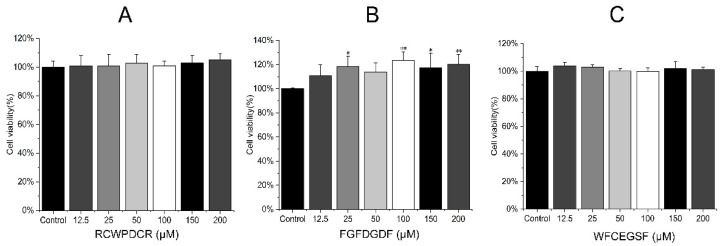
Effects of THPs on the viability of RIN-14B cells (*n* = 3). RIN-14B cells were pretreated with 0, 12.5, 25, 50, 100, and 200 μM of RCWPDCR (**A**), FGFDGDF (**B**), and WFCEGSF (**C**) for 24 h. * *p* < 0.05 vs. control; ** *p* < 0.01 vs. control.

**Figure 5 ijms-26-02119-f005:**
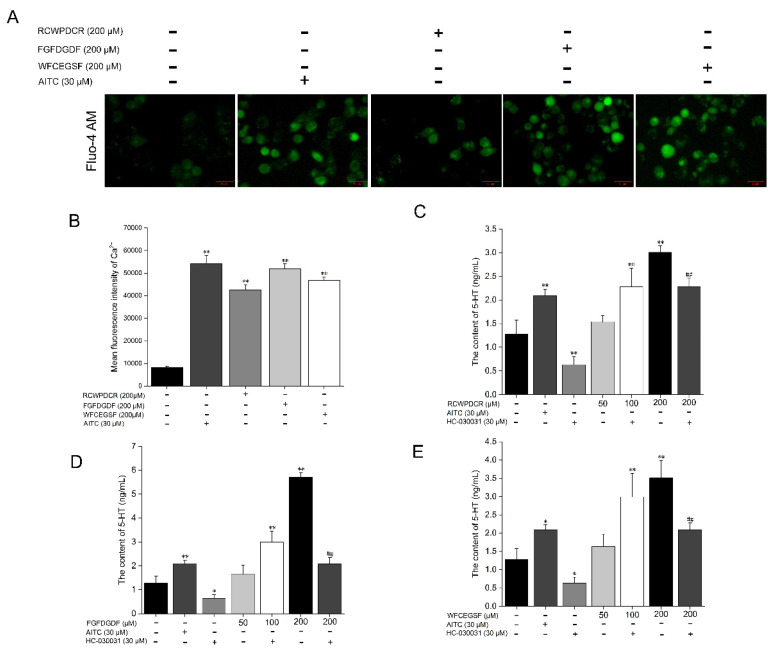
THP and AITC treatments of RIN-14B cells induce Ca^2+^ influx and 5-HT release and are positively modulated by the TRPA1 channel. (**A**,**B**) Fluorescence and its relative quantification for Ca^2+^ concentration upon THP and AITC treatments (*n* = 3). The release of 5-HT from RIN-14B cells with RCWPDCR (**C**), FGFDGDF (**D**), WFCEGSF (200 μM) (**E**), and AITC, a TRPA1 agonist (30 μM), along with the inhibition of the TRPA1 channel with HC-030031 (30 μM) (*n* = 3). Scale bar: 25 μm. Quantification data represented as mean ± SD. One-way ANOVA was applied for statistical analysis. * *p* < 0.05 vs. control; ** *p* < 0.01 vs. control; ^##^
*p* < 0.01 vs. 200 μM THPs.

**Figure 6 ijms-26-02119-f006:**
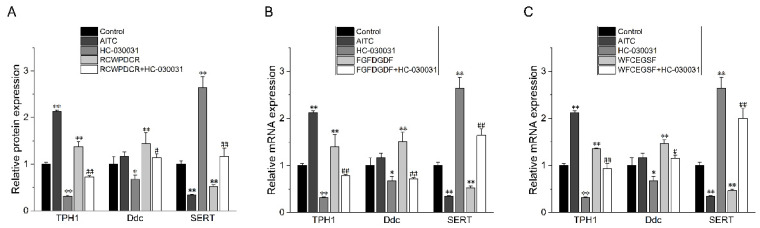
The RT-qPCR validation in the examination of the effects of the TRPA1 agonist AITC (30 μM), the TRPA1 inhibitor HC-030031 (30 μM), THPs (200 μM), and THPs (200 μM) + HC-030031 (30 μM) on the levels of the genes related to the 5-HT synthesis pathway, TPH1, SERT, and Ddc (*n* = 3). (**A**) RCWPDCR regulated the expression of the TPH1, Ddc, and SERT genes related to the 5-HT synthesis pathway. (**B**) FGFDGDF regulated the expression of the TPH1, Ddc and SERT genes related to the 5-HT synthesis pathway. (**C**) WFCEGSF regulated the expression of the TPH1, Ddc, and SERT genes related to the 5-HT synthesis pathway. * *p* < 0.05 vs. control; ** *p* < 0.01 vs. control; ^#^
*p* < 0.05 vs. 200 μM THPs; ^##^
*p* < 0.01 vs. 200 μM THPs.

**Figure 7 ijms-26-02119-f007:**
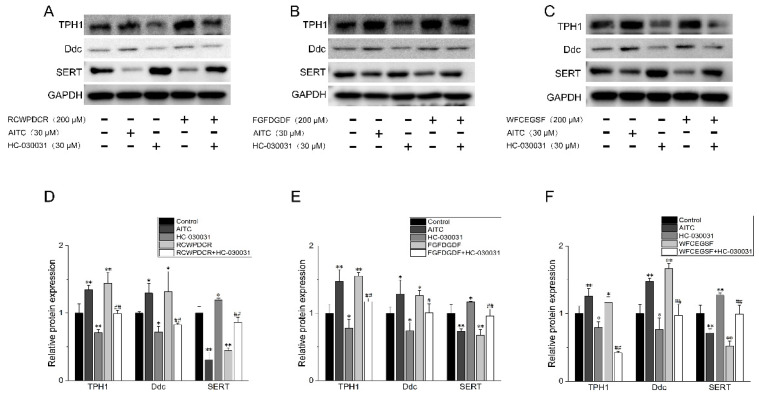
The Western blot validation in the examination of the effects of the TRPA1 agonist AITC (30 μM), the TRPA1 inhibitor HC-030031 (30 μM), THPs (200 μM), and THPs (200 μM) + HC-030031 (30 μM) on the levels of the genes related to the 5-HT synthesis pathway, TPH1, SERT, and Ddc (*n* = 3). (**A**,**D**) RCWPDCR: representative bands of a Western blot for TPH1, SERT, and Ddc and their relative protein expression. (**B**,**E**) FGFDGDF: representative bands of a Western blot for TPH1, SERT, and Ddc and their relative protein expression. (**C**,**F**) WFCEGSF: representative bands of a Western blot for TPH1, SERT, and Ddc and their relative protein expression. * *p* < 0.05 vs. control; ** *p* < 0.01 vs. control; ^#^
*p* < 0.05 vs. 200 μM THPs; ^##^
*p* < 0.01 vs. 200 μM THPs.

**Table 1 ijms-26-02119-t001:** Area, activity scores, the predictions of toxin, MW, and molecular docking scores of the 11 peptides from SAH.

Sequence	Area	PeptideRanker	Toxin Prediction	MW	ADCP Score/(kcal·mol^−1^)
RCWPDCR	6.18 × 10^6^	0.941094	Non-Toxin	935.16	−17.5
FGYY	1.36 × 10^6^	0.938	Non-Toxin	548.64	0
FGFDGDF	1.75 × 10^6^	0.93535	Non-Toxin	803.91	−22.6
WYLR	7.55 × 10^6^	0.904547	Non-Toxin	636.8	0
WFCEGSF	1.34 × 10^7^	0.88956	Non-Toxin	875.05	−21.5
TYFPFH	1.34 × 10^7^	0.880767	Non-Toxin	875.05	−15.3
LYYAPF	2.69 × 10^6^	0.878367	Non-Toxin	772.97	−15.8
GPSGPQGPSGPLQGP	8.65 × 10^6^	0.838269	Non-Toxin	1332.66	0
MLCVGFL	4.03 × 10^7^	0.8193	Non-Toxin	782.13	−17.4
VPTCF	1.69 × 10^6^	0.815237	Non-Toxin	565.74	−12.3
MLCVGF	4.03 × 10^7^	0.811293	Non-Toxin	668.95	−15.7

**Table 2 ijms-26-02119-t002:** MM/GBSA (kcal/mol) predicted free energy binding and energy components.

	FGFDGDF-TRPA1	RCWPDCR-TRPA1	WFCEGSF-TRPA1
Δ_VDWAALS_	−48.70 ± 0.03	−59.30 ± 0.15	−59.78 ± 0.45
Δ_EEL_	−751.47 ± 1.66	−421.09 ± 0.39	−422.08 ± 8.66
Δ_EGB_	766.96 ± 3.58	434.12 ± 5.46	461.56 ± 19.08
Δ_ESURF_	−9.25 ± 0.05	−11.16 ± 0.05	−9.36 ± 0.43
Δ_GGAS_	−800.17 ± 1.66	−480.40 ± 0.41	−481.86 ± 8.68
Δ_GSOLV_	757.71 ± 3.58	422.96 ± 5.46	452.20 ± 19.09
Δ_TOTAL_	−42.46 ± 3.95	−57.44 ± 5.47	−29.66 ± 20.97

**Table 3 ijms-26-02119-t003:** Primer sequence.

Genes	Primer Sequence (5′-3′)	Primer Length (bp)
TPH1	Forward: GACTGCGACATCAACCGAGAAC	22
	Reverse: CGGGCGAGTCCACAGAGAG	19
SERT	Forward: GCGGAGATGAGGAATGAAGATGTG	24
	Reverse: CGTGGATGCTGGCATGTTGG	20
Ddc	Forward: ACTTGGTTCCGTGTCGTCTC	20
	Reverse: CTTTCTCTGCCCTCAGCACA	20
GAPDH	Forward: CCCTTAAGAGGGATGCTGCC	20
	Reverse: TACGGCCAAATCCGTTCACA	20

## Data Availability

Data will be made available upon request.

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
