# Peer review of "TRPA1-Activated Peptides from Saiga Antelope Horn: Screening, Interaction Mechanism, and Bioactivity"

_ijms, 2025, doi:10.3390/ijms26052119_

Round 1

Reviewer 1 Report

Comments and Suggestions for Authors

The manuscript reported 3 bioactive peptides (RCWPDCR, FGFDGDF and WFCEGSF) screened from Saiga antelope horn. These peptides are described as having high affinity for the transient receptor potential ankyrin 1 (TRPA1) channel and the ability to enhance 5-HT release.

1. In the paper, the authors claim that the screened peptides specifically bind to TRRA1 by ultrafiltration. How was it verified that these peptides do not exhibit high affinity for other receptors or proteins?

2. The introduction is not well-structured. The paper mentions that saiga antelope derivatives are traditionally used to treat febrile convulsions, fever, eclampsia, and hemorrhagic disorders, and that TRPA1 is associated with 5-HT release, which in turn is linked to neurological and metabolic disorders. However, the manuscript does not clearly identify the potential diseases that could be treated using TRPA1-specific high-affinity peptides. The authors should summarize this connection in the introduction and emphasize the significance of developing these peptides to address specific medical conditions.

3. The manuscript contains several unclear expressions. For example, on Line 66, “TRPA1 channels are closely related to 5-HT secretion” and “TRPA1 has important roles in the pathophysiology of almost all organ systems” lack sufficient scientific detail. The authors should provide more precise and scientifically supported explanations to summarize the relationship between TRPA1 and 5-HT secretion, as well as its significance in various organ systems.

4. In 2.3, after incubation, what solution have been used for washing to remove the unspecific binding?

5. In Figure 1D, there are additional peptides derived from affinity ultrafiltration (blue color) that are not included among the peptides derived from the antelope horn. Where are these peptides originally from?

6. The unit should be consistent. Line 147 and 149, “microliters” or “μL”. And “μM” in the figure and “μmol/L” in the method parts.

7. In figure 3, the statistical analysis should be reconducted by ANOVA.

8. For molecular docking analysis, which docking mold was used in the study? Flexible docking, semi-flexible or rigid docking? Additionally, what are the major interaction forces between the peptides and TRPA1? What are the specific sites in TRPA1 structure defined for activation or inhibition?

9. The overall resolution of the figures needs to be improved. In Figure 1 and Figure 2,  it is very difficult to recognize the number in the figures. 

Overall, this manuscript is not recommended for publication in its current form.

Comments on the Quality of English Language

 The English should be improved to more clearly express the research.

Author Response

Comments 1: In the paper, the authors claim that the screened peptides specifically bind to TRRA1 by ultrafiltration. How was it verified that these peptides do not exhibit high affinity for other receptors or proteins?

Response 1: Thank you very much for your careful review and valuable comments on our manuscript. We very much appreciate the questions you have raised, which is indeed a very important and worthwhile aspect to be explored in depth. As you are concerned, we agree that more studies would be useful to understand the details of interaction and enhancement. Our study focuses on answering critical questions regarding the TRPA1 activated peptides from Saiga antelope horn. In this study, we have ensured a high affinity for the TRPA1 protein by ultrafiltration affinity screening of the peptide. And its specificity on TRPA1 protein has been further demonstrated by molecular docking and cellular experiments. However, it is true that we did not perform affinity validation on other non-target receptors or proteins, as mentioned by the reviewer. Therefore, we plan to further confirm whether these peptides exhibit high affinity only for TRPA1 and low affinity for other receptors or proteins using, for example, competitive binding assays or MicroScale-Thermophoresis (MST) techniques in subsequent studies. Thank you again for asking this valuable question, your careful review has greatly enhanced the quality of our paper.

Comments 2: The introduction is not well-structured. The paper mentions that saiga antelope derivatives are traditionally used to treat febrile convulsions, fever, eclampsia, and hemorrhagic disorders, and that TRPA1 is associated with 5-HT release, which in turn is linked to neurological and metabolic disorders. However, the manuscript does not clearly identify the potential diseases that could be treated using TRPA1-specific high-affinity peptides. The authors should summarize this connection in the introduction and emphasize the significance of developing these peptides to address specific medical conditions.

Response 2: Thank you very much for your careful review and valuable suggestions on our manuscript. We greatly appreciate the questions you raised, which is indeed a very important point that can help us better clarify the structure and context of the paper. In the revised manuscript, we have revised the Introduction as follows: “In summary, we hypothesize that a large number of peptides produced by antelope horn after gastrointestinal digestion can exert antipyretic effects by up-regulating the expression of 5-HT through TRPA1 protein on enterochromaffin cells. However, determining the distinctive peptides of Antelope horn with unique potency is a pressing task. TRPA1-specific high-affinity peptides (TSHPs) were discovered from the degraded peptide mixture of antelope horn utilizing AUF-LC/MS, bioactivity databases, and molecular docking approaches. Furthermore, the mechanism of TSHPs in controlling serotonin release from RIN-14B cells was investigated, revealing new information about the mechanism of action of bioactive peptides from SAH in suppressing FS. Furthermore, this work lends scientific support to the creation of artificial alternatives to SAH.” We believe that these modifications will better highlight the importance of the study and enhance the logic of the article. We hope that these modifications will fulfill your requirements. Thank you again for your valuable suggestions, your careful review has greatly enhanced the quality and logical structure of our article.

Comments 3: The manuscript contains several unclear expressions. For example, on Line 66, “TRPA1 channels are closely related to 5-HT secretion” and “TRPA1 has important roles in the pathophysiology of almost all organ systems” lack sufficient scientific detail. The authors should provide more precise and scientifically supported explanations to summarize the relationship between TRPA1 and 5-HT secretion, as well as its significance in various organ systems.

Response 3: Thank you very much for your careful review of our manuscript and your valuable suggestions. We deeply appreciate the questions you asked, and your demand for precision of expression and scientific basis has provided us with excellent guidance and helped us to improve the quality of our paper. For the sentences “TRPA1 channels are closely related to 5-HT secretion” and “TRPA1 plays an important role in the pathophysiology of almost all organ systems”, we have revised them and added more specific scientific details in revised manuscript. Here is the revised version:“Sodium oligomannate has been shown to either directly or indirectly enhance calcium entry in enteroendocrine cells by activating sweet taste receptors and TRPA1. This leads to an increase in CCK and 5-HT release, which in turn increases vagal afferent activity. Sodium oligomannate may influence cognitive processes by activating the ECs-vagal afferent pathways[8]. Additionally, hydrogen sulfide activates TRPA1 and promotes the release of 5-HT from epithelioid cells of the chicken thoracic aorta[9].” “Mutations in the TRPA1 gene may not only impact sensory nerves and microvasculature, causing neurological discomfort and vascular issues[12] , but may also contribute to ozone-induced lung injury[13]. Wang et al. discovered that TRPA1 can prevent con-trast-induced acute kidney injury by controlling mitochondrial fission/fusion, biogene-sis, and dysfunction. TRPA1 activation may provide a novel treatment strategy for pre-venting contrast-induced acute renal damage[14].”we believe that these changes will make the statements more accurate and scientific. Thank you again for your invaluable comments, your careful review and insights have greatly helped us to refine the content of the manuscript.

Comments 4: In 2.3, after incubation, what solution have been used for washing to remove the unspecific binding?

Response 4: Thank you very much for your careful review of our manuscript and for your questions. We appreciate your attention to experimental details. Regarding your question in section 2.3, “what solution have been used for washing to remove the unspecific binding”, we have already described it in the original manuscript. Specifically, we used 90% methanol-water as the washing solution to remove non-specific binding. It is possible that the expression in the text was not prominent enough to draw your attention, so we will further emphasize this part in the revised manuscript to ensure that the information is clearer. Thank you again for your valuable comments, your careful review helped us tremendously to improve the quality of the article.

Comments 5: In Figure 1D, there are additional peptides derived from affinity ultrafiltration (blue color) that are not included among the peptides derived from the antelope horn. Where are these peptides originally from?

Response 5: Thank you very much for your careful review of our manuscript and your valuable suggestions. Your questions were very precise and helped us to think more clearly about what is in the charts and make improvements. The peptides shown in Figure 1D (blue color) are not derived from antelope horn derivatives, but are peptides produced during incubation and elution that do not bind to TRPA1 protein. These peptides had no other role in our study, so they were not elaborated. Thank you again for your careful review, your questions greatly helped us to clarify the expression of this section and improve the accuracy of the paper.

Comments 6: The unit should be consistent. Line 147 and 149, “microliters” or “μL”. And “μM” in the figure and “μmol/L” in the method parts.

Response 6: Thank you very much for your careful review and valuable comments on our manuscript. We very much appreciate your concern about the consistency of units, which is crucial for the accuracy of the article. Regarding your question on “units should be consistent”, there is indeed an inconsistency in the use of 'μL' and 'μM' in lines 147 and 149, and we have revised these sections to ensure the consistency of units throughout the article. these sections to ensure consistency of units across the text. Specifically, we have standardized the units to 'μL' and 'μM' and have made corresponding adjustments in the figures. Thank you again for your careful review and suggestions, which greatly helped us to improve the quality of the article.

Comments 7: In figure 3, the statistical analysis should be reconducted by ANOVA.

Response 7: Thank you very much for your careful review of our manuscript and your valuable suggestions. We greatly appreciate your interest in statistical analysis and your suggestions are important to ensure rigor and accuracy in data analysis. We note that there is indeed a presentation error in the manuscript, as we wrote “one-way analyses of variance” in the original text, but we actually used the correct name of the statistical method, one-way ANOVA. We have revised the manuscript and corrected this description. We appreciate your careful review and have corrected the statistical method in 2.14 of the manuscript to ensure that the description in the text is consistent with the actual analysis. Thank you again for your careful review and valuable comments, your professional advice has greatly helped us to improve the science and accuracy of the article.

Comments 8: For molecular docking analysis, which docking mold was used in the study? Flexible docking, semi-flexible or rigid docking? Additionally, what are the major interaction forces between the peptides and TRPA1? What are the specific sites in TRPA1 structure defined for activation or inhibition?

Response 8: Thank you very much for your careful review and valuable suggestions on our manuscript. Your questions about molecular docking analysis were very important and precise, helping us to further clarify the details of the experimental design and analysis. Your questions were very constructive and prompted us to revisit the descriptions in the relevant sections. In our study, we used the rigid docking method. Specifically, we docked peptides and proteins using ADCP. ADCP, known as AutoDock CrankPep, is a module inside the AuckDock program dedicated to peptide docking. It combines protein folding techniques as well as the effective form of rigid receptors as the basis for affinity calculations, and performs operations on peptide folding in the context of the energy landscape of the receptor. While optimizing the interaction between the peptide and the receptor protein, ADCP employs Monte Carlo methods to compute the folding of the peptide to produce a docked conformation of the peptide.

Regarding the main interaction forces between the peptide and TRPA1, the analysis results showed that the peptide interacts with TRPA1 mainly through conventional hydrogen bonding, Pi-cation, Pi-alkyl and carbon-hydrogen bonding with the active binding site of TRPA1.

Furthermore, with respect to the specific sites in the TRPA1 structure used for activation or inhibition, the electrophilic ligands are able to activate the TRPA1 channel through covalent modification and/or disulfide bonding by interacting with key cysteine residues at the N-terminal end of the channel. The activation of electrophilic compounds depends on their thiol-responsive part, taking into account the structural diversity of this organization. On the other hand, the non-electrophilic ligand does not interact with the key cysteine on the channel, so the structural diversity of this motif is not yet explained.

We have elaborated on these issues in the revised manuscript and further improved the molecular docking analysis section based on your suggestions. Thank you again for your valuable questions, your careful review helped us to express and elaborate these important scientific details more clearly. Your suggestions have greatly enhanced the quality and accuracy of the paper.

Comments 9: The overall resolution of the figures needs to be improved. In Figure 1 and Figure 2, it is very difficult to recognize the number in the figures.

Response 9: Thank you very much for your careful review of our manuscript and your valuable suggestions. We greatly appreciate your attention to the clarity of the graphs and charts, which is essential to ensure that readers are able to understand the data accurately. We have noticed that the resolution of the numbers in the figures is indeed low, making the numbers in the images illegible. The images have been replaced with higher resolution images, and to avoid compression of the image quality by the submission system during the submission process, we have uploaded each image individually for reviewers to download and review. Thank you again for your valuable comments, your careful review helped us to further improve the quality and readability of the manuscript.

4. Response to Comments on the Quality of English Language

Point 1: The English should be improved to more clearly express the research.

Response 1: Thanks for your suggestion. We feel sorry for our poor writings. We tried our best to improve the manuscript and made some changes to the manuscript. These changes will not influence the content and framework of the paper. And here we did not list the changes but marked in red in the revised paper. We appreciate for Reviewers’ warm work earnestly and hope that the correction will meet with approval.

5. Additional clarifications

We tried our best to improve the manuscript and made some changes marked in red in the revised paper, which will not influence the content and framework of the paper. We appreciate the reviewers warm work earnestly and hope the correction will meet with approval. Once again, thank you very much for your comments and suggestions.

Reviewer 2 Report

Comments and Suggestions for Authors

Firstly I would like to express my concerns about the sources of SAH. The authors in the introduction part they write.. The wild Saiga antelope population has significantly decreased as a result of the SAH trade's explosive expansion in recent decades, and the usage of SAH is currently prohibited

I am just wondering... If the usage of SAH is prohibited how the company sells it? Is it stock from the previous years? It is quite strange this.. Anyway... 

The study contributes to understanding peptide-based TRPA1 activation and potential replacements for SAH, which are important for conservation efforts.

The experimental design is robust, utilizing affinity ultrafiltration, LC-MS/MS, molecular docking, and cellular assays. 

In the abstract part there is a phrase... The phrase "high-efficiency activation" is vague. Please revision it to connect better the experimental results.

The introduction provides good background information on Saiga antelope horn (SAH) as a traditional medicine and its pharmacological relevance but could further clarify how its components are linked to TRPA1 activation.

In the paragraph 2.5 the authors used some online models... while they cite them in the bibliography in the paragraph the numbers are missing. Please revision it

In the next paragraph molecular docking is presented. However molecular docking if often insufficient to confirm strong binding interactions. Molecular dynamics simulations are rather necessary and would improve credibility.

Peptides are bought from a peptide company so there is no point to have paragraph 2.7.... 

In the discussion part.. the authors write... However, the main components of SAH, proteins and peptides, rarely cross the blood-brain barrier to exert pharmacological effects.  Did the authors checked with other online models if SAH peptides can cross BBB... the findings of this study would benefit the article and in general the knowledge for SAH peptides.

The study lacks direct comparisons with known TRPA1 activators, making it difficult to assess the peptides’ effectiveness. It would be better to compare them with them.

To finalize my comments I think the authors should perform the following:

Address ethical sourcing concerns of SAH

Perform molecular dynamics simulations for docking validation.

And lastly, include benchmarking against known TRPA1 activators

These significant revisions are necessary and they will add high value to the article.

Author Response

Comments 1: Firstly I would like to express my concerns about the sources of SAH. The authors in the introduction part they write.. The wild Saiga antelope population has significantly decreased as a result of the SAH trade's explosive expansion in recent decades, and the usage of SAH is currently prohibited I am just wondering... If the usage of SAH is prohibited how the company sells it? Is it stock from the previous years?

Response 1: Thank you very much for your careful review of our manuscript and your valuable suggestions. We are deeply impressed by the insightful questions you asked, and your concerns show your in-depth understanding of the content of the article, which greatly helped us to refine the details in the article. We understand your concerns about the statement “SAH use is currently banned”. In fact, despite the recent ban due to the decline of the Saiga antelope population, the SAH derivatives used in our study were sourced from legal stocks of materials that were legally obtained before the ban was imposed.

In order to comply with the regulations, we have ensured that all SAH derivatives used meet current legal requirements and are used for scientific purposes only. We have updated the relevant statements in the manuscript to more clearly articulate the origin and legality of these materials. Thank you again for your very valuable questions, and your careful review has helped us to more accurately represent the contextual information in the paper and improve the rigor of the article.

Comments 2: In the abstract part there is a phrase... The phrase "high-efficiency activation" is vague. Please revision it to connect better the experimental results.

Response 2: Thank you very much for your careful review of our manuscript and your valuable suggestions. We are deeply impressed by your precise and constructive questions, and your attention to the presentation of the abstract has helped us to further improve the clarity and rigor of the article. We note that the expression “high-efficiency activation” is indeed vague. In order to better relate it to the experimental results, we have modified the expression. By deleting “high-efficiency” we believe that this change will make the abstract more accurate and better reflect the experimental results. Thank you again for your very valuable comments, your careful review has greatly helped us to refine the presentation of the article for clarity and precision.

Comments 3: The introduction provides good background information on Saiga antelope horn (SAH) as a traditional medicine and its pharmacological relevance but could further clarify how its components are linked to TRPA1 activation.

Response 3: Thank you very much for your careful review of our manuscript and valuable suggestions. We greatly appreciate your comments and your suggestions for further elaboration of the relationship between Saiga antelope horn (SAH) components and TRPA1 activation are very valuable and help us to better clarify the scientific context in the article. We agree that how the components of SAH are associated with TRPA1 activation could be further elaborated in this section to provide readers with a clearer understanding. According to this paper, some peptides in SAH may induce activation of the TRPA1 receptor by binding to it. We have further detailed in the manuscript how peptides, major components of SAH, affect TRPA1 receptor function through specific molecular mechanisms by ultrafiltration affinity, molecular docking and cellular analysis. We have revised the introductory section based on your suggestion to make this point clearer and cited relevant literature to support the thesis. Thank you again for your very valuable questions, your careful review has helped us immensely and enhanced the depth of the article!

Comments 4: In the paragraph 2.5 the authors used some online models... while they cite them in the bibliography in the paragraph the numbers are missing. Please revision it

Response 4: Thank you very much for your careful review of our manuscript and your valuable suggestions. We are deeply impressed by the importance of the issues you raised, and your careful review has helped us to identify omissions in citation and numerical labeling, which has greatly improved the accuracy and completeness of the article. We note that although we have cited the relevant online models in the paragraph, the corresponding figures are indeed missing. We have amended the manuscript according to your suggestion to ensure that all cited models and figures are correctly labeled and referenced. Thank you again for your very valuable questions; your careful review enabled us to better refine the article and enhance its academic rigor.

Comments 5: In the next paragraph molecular docking is presented. However molecular docking if often insufficient to confirm strong binding interactions. Molecular dynamics simulations are rather necessary and would improve credibility.

Response 5: Thank you very much for your careful review and valuable suggestions on our manuscript. We are deeply impressed by the depth of the questions you raised, and your professional comments provide important directions for us to further improve the paper, especially in terms of the credibility of the molecular docking analysis. Therefore, we have supplemented the manuscript with molecular dynamics simulation experiments to further validate the stability and kinetic features of binding. Through molecular dynamics simulations, we were able to gain a deeper understanding of TSHPs-TRPA1 interaction, especially in terms of binding stability and conformational changes. These results enhance our confidence in molecular docking predictions and provide more reliable data support. We describe the experimental design and results of the molecular dynamics simulations in detail in sections 2.7 and 3.3 of the manuscript to ensure that this section is well presented. Thank you again for your very valuable suggestions. Your careful review and professional advice have greatly helped us to improve the quality and scientific rigor of the article.

Comments 6: Peptides are bought from a peptide company so there is no point to have paragraph 2.7

Response 6: Thank you very much for your careful review of our manuscript and valuable suggestions. We greatly appreciate the questions you asked and your attention to the details of the article helped us to further streamline and improve the clarity of the article. We understand that the peptides were purchased from the peptide company and that no complex customization or synthesis was performed. Therefore, there is really no need to describe the paragraph in detail. Following your suggestion, we have deleted paragraph 2.7 to simplify the manuscript and avoid redundancy. Thank you again for your valuable suggestions. Your careful review enabled us to optimize the structure of the article and enhance its overall rigor and flow.

Comments 7: In the discussion part. the authors write... However, the main components of SAH, proteins and peptides, rarely cross the blood-brain barrier to exert pharmacological effects.  Did the authors checked with other online models if SAH peptides can cross BBB... the findings of this study would benefit the article and in general the knowledge for SAH peptides.

Response 7: Thank you very much for your careful review of our manuscript and your valuable suggestions. We greatly appreciate your questions, which were very insightful and helped us to further improve the scientific quality and completeness of the article. We apologize for the lack of rigor in the statement “However, the main components of SAH, proteins and peptides, rarely cross the blood-brain barrier to exert pharmacological effects.”, and for the lack of sufficient consideration of existing studies. We have corrected this statement in the revised manuscript to discuss the blood-brain barrier penetration of peptides more critically. The corrections are as follows:” In recent years, it has been found that the degraded peptide fragments of keratin-like components in horned animal medicines may be the important efficacy material basis of horned animal medicines. Antelope horn is directly used as powder in medicine, and its main component keratin is degraded into a large number of peptide compounds after gastrointestinal digestion. It is well known that the special biochemical environment of the gastrointestinal tract, mucus, epithelial permeation and liver elimination and other harsh conditions greatly restrict the peptides from being absorbed into the blood circulation through the intestines. More importantly, it is difficult for most com-pounds in the blood circulation, including peptides, to penetrate the blood-brain barrier and enter the brain. Therefore, we hypothesized that SAH peptide could activate intestinal TRPA1 channels and increase 5-HT secretion to treat FS.” Thank you again for your very valuable suggestions. Your careful review and professional advice have greatly helped us to improve the quality and scientific rigor of the article.

Comments 8: The study lacks direct comparisons with known TRPA1 activators, making it difficult to assess the peptides’ effectiveness. It would be better to compare them with them.

Response 8: Thank you very much for your careful review of our manuscript and valuable suggestions. We greatly appreciate your questions and your very insightful comments on the comparative analysis with known TRPA1 activators, which is indeed crucial for assessing peptide efficacy. Regarding your reference to the lack of direct comparison with known TRPA1 activators, we have actually performed relevant experiments in our study and compared the effects of the peptides we screened with known TRPA1 activators——Allyl isothiocyanate (AITC) in the original manuscript. It is possible that the lack of clarity in the original manuscript prevented this from being clearly communicated to you. In the revised manuscript, we have provided a more detailed description of the experimental results to ensure that this comparative analysis is more clearly visible. We have explained AITC in the Material in the revised manuscript and labeled AITC in the figure legends. Thank you again for your valuable suggestions. Your careful review has helped us to further improve the presentation and scientific rigor of the article.

4. Additional clarifications

We tried our best to improve the manuscript and made some changes marked in red in the revised paper, which will not influence the content and framework of the paper. We appreciate the reviewers warm work earnestly and hope the correction will meet with approval. Once again, thank you very much for your comments and suggestions.

Round 2

Reviewer 1 Report

Comments and Suggestions for Authors
  1. The authors discuss the specific binding between the peptide and TRPA1, however, there is no direct evidence or experimental design provided to support this claim. Instead of referring to "specific binding," it would be more appropriate to describe it as a high-affinity interaction between the peptide and TRPA1.
  2. The resolution of some figures is still very low, such as figure 1b, figure 1c, figure 2a-f. It must be improved to make the figure readable.
Comments on the Quality of English Language

No.

Author Response

Comments 1: The authors discuss the specific binding between the peptide and TRPA1, however, there is no direct evidence or experimental design provided to support this claim. Instead of referring to "specific binding," it would be more appropriate to describe it as a high-affinity interaction between the peptide and TRPA1.

Response 1: Thank you very much for your careful review of our manuscript and valuable suggestions. We appreciate your concern about the specific binding of peptides to TRPA1 in the article, and your questions were very constructive and helped us to improve the scientific rigor of the article. Regarding your reference to the use of the term “specific binding”, we fully understand your point of view. We did describe the binding between the peptide and TRPA1 in the article, but as you pointed out, we were unable to provide direct experimental data or evidence to fully support this claim. Therefore, in the revised manuscript we will change the expression to use “high-affinity” instead of “specific binding”. This will more accurately reflect the relationship between the peptide and TRPA1 and ensure a more scientific and rigorous presentation. Thank you again for your very valuable suggestions, and your careful review has greatly helped us to refine the presentation of the article to make it more precise and credible.

Comments 2: The resolution of some figures is still very low, such as figure 1b, figure 1c, figure 2a-f. It must be improved to make the figure readable.

Response 2: Thank you very much for your careful review of our manuscript and your valuable suggestions. We greatly appreciate your concern about the quality of the figures and your questions are very helpful in enhancing the readability and professionalism of the article. Regarding your mention of the low resolution of Figures 1b, 1c, and 2a-f, we fully understand your concern. Indeed, the clarity and resolution of the images are critical to the presentation of the data. In the revised manuscript, we have reprocessed these figures and increased the resolution to ensure they are more legible. Thank you again for your valuable comments, your careful review has greatly helped us to improve the quality of the article and the readability of the charts.

4. Response to Comments on the Quality of English Language

Point 1: The English should be improved to more clearly express the research.

Response 1: Thanks for your suggestion. We feel sorry for our poor writings. We tried our best to improve the manuscript and made some changes to the manuscript. These changes will not influence the content and framework of the paper. And here we did not list the changes but marked in red in the revised manuscript. We appreciate for Reviewers’ warm work earnestly and hope that the correction will meet with approval.

Reviewer 2 Report

Comments and Suggestions for Authors

Thanks for your revisions and the responses

Author Response

Thank you very much for your positive comments and suggestions to our manuscript.